# The assessment of physiotherapy practice is a robust measure of entry-level physiotherapy standards: Reliability and validity evidence from a large, representative sample

Alan Reubenson[1*☯], Leo Ng[1,2], Vidya Lawton[3], Irmina Nahon[4], Rebecca Terry[5], Claire Baldwin[6], Julia Blackford[7], Alex Bond[6], Rosemary Corrigan[8], Megan Dalton[9,2], Amabile Borges Dario[7], Michael Donovan[10], Ruth Dunwoodie[10], Genevieve M. Dwyer[11], Roma Forbes[10], Alison Francis-Cracknell[12], Janelle Gill[13], Andrea Hams[14], Anne Jones[15], Taryn Jones[14], Belinda Judd[7], Ewan Kennedy[16], Prue Morgan[12], Tanya Palmer[17], Casey Peiris[18], Carolyn Taylor[19], Debra Virtue[20], Cherie Zischke[8], Daniel F. Gucciardi[1☯], on behalf of the Physiotherapy Clinical Education Research Collaborative (PCERC)[¶]

1 Curtin School of Allied Health, Curtin University, Perth, Australia, 2 School of Health Sciences, Swinburne University of Technology, Melbourne, Australia, 3 Faculty of Medicine, Health and Human Sciences, Macquarie University, Sydney, Australia, 4 School of Rehabilitation and Exercise Sciences, University of Canberra, Canberra, Australia, 5 Department of Physiotherapy, Faculty of Health Sciences and Medicine, Bond University, Gold Coast, Australia, 6 College of Nursing and Health Sciences and Caring Futures Institute, Flinders University, Adelaide, Australia, 7 School of Health Sciences, The University of Sydney, Sydney, Australia, 8 School of Allied Health, Exercise and Sports Sciences, Charles Sturt University, Australia, 9 School of Allied Health, Australian Catholic University, Brisbane, Australia, 10 School of Health and Rehabilitation Sciences, The University of Queensland, Brisbane, Australia, 11 School of Health Sciences, Western Sydney University, Sydney, Australia, 12 Department of Physiotherapy, Monash University, Melbourne, Australia, 13 School of Health Sciences, The University of Notre Dame Australia, Fremantle, Australia, 14 School of Health Sciences and Social Work, Griffith University, Gold Coast, Australia, 15 College of Healthcare Sciences, James Cook University, Townsville, Australia, 16 Centre for Health, Activity and Rehabilitation Research, School of Physiotherapy, University of Otago, Dunedin, New Zealand, 17 School of Health, Medical and Applied Science, Central Queensland University, Bundaberg, Australia, 18 Academic and Research Collaborative in Health (ARCH), La Trobe University and the Royal Melbourne Hospital, Melbourne, Australia, 19 La Trobe Rural Health School, La Trobe University, Bendigo, Australia, 20 School of Health Sciences, The University of Melbourne, Melbourne, Australia,

¶ Membership of the Physiotherapy Clinical Education Research Collaborative is provided in the acknowledgements.
☯ These authors contributed equally to this work.
* A.Reubenson@curtin.edu.au

## Abstract

The Assessment of Physiotherapy Practice (APP) is a 20-item assessment instrument used to assess entry-level physiotherapy practice in Australia, New Zealand and other international locations. Initial APP reliability and validity evidence supported a unidimensional or single latent factor as the best representation of entry-level physiotherapy practice performance. However, there remains inconsistency in how the APP is interpreted and operationalised across Australian and New Zealand universities offering entry-level physiotherapy programs. In essence, the presumption that the psychometric integrity of the APP

**Data availability statement:** The participants of this study did not provide written consent for their data to be shared publicly. Due to the sensitive nature of the data, namely student academic records, supporting data is unavailable. Data are available on request from the Curtin University Human Research Ethics Committee (hrec@curtin.edu.au) for researchers who meet the criteria to gain access to confidential data.

**Funding:** The author(s) received no specific funding for this work.

**Competing interests:** NO authors have competing interests.

generalises across people, time, and contexts remains largely untested. This multi-site, archival replication study utilised APP assessment data from 8,979 clinical placement assessments, across 19 Australian and New Zealand universities, graduating entry-level physiotherapy students (n=1865) in 2019. Structural representation of APP scores were examined via confirmatory factor analysis and penalised structural equation models. Factor analyses indicated a 2-factor representation, with four items (1–4) for the professional dimension and 16 items (5–20) for the clinical dimension, is the best approximation of entry-level physiotherapy performance. Measurement invariance analyses supported the robustness of this 2-factor representation over time and across diverse practice areas in both penultimate and final years of study. The findings provide strong evidence for the psychometric integrity of the APP, and the 2-factor alternative interpretation and operationalisation is recommended. To meet entry-level standards students should be assessed as competent across both professional and clinical dimensions of physiotherapy practice.

## Introduction

Clinical placements are integral to the training pathways of physiotherapists and other healthcare professionals, providing students with opportunities to integrate knowledge and skills obtained from campus-based learning into real-world settings [1]. Scientifically robust assessment of performance is important to ensure graduates demonstrate the required professional standards necessary for safe and effective healthcare [2]. The Assessment of Physiotherapy Practice (APP) was developed as a standardised instrument, within Australia and New Zealand (ANZ), for the assessment and evaluation of physiotherapy student performance against the same entry-level standards across diverse contextual considerations [3–5]. The APP is the preferred student performance assessment instrument among entry-level physiotherapy programs across ANZ, yet its implementation varies around the interpretation of passing criteria [3–11] and psychometric evidence would benefit from continued investigation [7,12]. Specifically, there is limited research regarding the extent to which the psychometric integrity of the APP generalises across people, time, and context. Relatedly, context of physiotherapy practice has diversified considerably since the inception of the APP 15 years ago with expansion in sectors such as disability and aged care [13,14].

Pre-registration physiotherapy education programs require students to meet 'entry-level standards' for registration as a physiotherapist. By the end of the physiotherapy degree, graduates need to have demonstrated the necessary knowledge, skills, attitudes and attributes to practise as autonomous, safe and ethical physiotherapy practitioners in the country they studied [15,16]. Within ANZ, the respective Physiotherapy Boards have collaborated to define the entry-level competencies and practice thresholds required for registration [17]. These competencies include the ability to operate as primary contact practitioners and cover seven key roles: physiotherapy practitioner; professional and ethical practitioner; communicator; reflective practitioner and self-directed learner; collaborative practitioner; educator; and manager/leader [17]. To meet these standards, pre-registration educational curricula will vary, yet they must include professional practice, in the form of clinical placements, in a range of areas that physiotherapists work [16].

The workplace learning environment is complex and dynamic, relying on active participation and social interactions to enhance learning and development [18–20]. In these settings, students can test and apply previously acquired skills and knowledge. Under the supervision of registered physiotherapists, students provide care to a diverse range of patients and clients

[21]. The authentic workplace environment can be a powerful and transformative learning space for students when all aspects such as practical experience, mentorship, and collaboration are considered holistically and alongside application of academic knowledge with real-world application [22]. The multidimensional nature of workplace learning can be considered using a practice development crucible metaphor, which illustrates how various considerations interact to shape student learning experiences [20]. This metaphor encompasses four key intersecting influences: (1) workplace influences, which include physical resources and workplace culture; (2) engagement in professional practices, highlighting the importance of student interactions with patients and staff; (3) clinical supervisors' intentions and actions, reflecting how supervisors' approaches influence student learning; and (4) students' dispositions and experiences, which encompass students' confidence and motivation. Understanding these influences is essential for maximising student learning in clinical placements. Students' dispositions and prior experiences including their knowledge, skills, and self-reflection, as well as characteristics such as confidence and engagement within the multi-disciplinary team play a crucial role in their learning. Nevertheless, these independent influences rarely occur in isolation and should be understood within the broader context of workplace elements, such as the culture and practices of the workplace, as well as the actions and intentions of clinical supervisors, like the quality of feedback they provide and their motivation to support student learning [23]. Therefore, evaluating student performance in these environments requires a comprehensive approach that considers all these interconnected elements.

Assessment of student physiotherapy practice performance is a complex endeavour that has evolved substantially over the past half century [24]. In the health sciences, work-based assessments are now the primary method for evaluating performance in real-world practice settings through direct observation [25,26]. In these contexts, assessment processes are influenced by the social and cultural contexts in which they occur, including interactions between students, supervisors, and the broader healthcare environment, all of which shape feedback and learning experiences. Work-based assessment within this framework relies on both summative and formative feedback processes to determine the achievement of professional standards [24,26]. A multitude of physiotherapy work-based performance assessment tools exist, yet most come with poorly reported psychometric properties, highlighting the need for ongoing psychometric work to improve the rigour of professional practice assessment processes [12]. In the physiotherapy discipline, evaluation of student performance commonly utilises a longitudinal assessment process, whereby student performance is observed repeatedly over a period of 4–6 weeks, by one or more clinical supervisors. This assessment process allows for variation in performance over time and under different circumstances and contexts [26]. It also acknowledges that workplaces and healthcare often vary, particularly regarding considerations like culture, policies, and processes as well as the relational interactions with supervisors and patients [18,26]. It is widely accepted that reliance on psychometric criteria alone fails to capture the complex social, cultural, and environmental interactions occurring in real-world practice settings and that performance-based assessment needs to involve a holistic, whole system approach [24,26–28]. Nevertheless, in a professional discipline like physiotherapy where there are defined threshold and competency standards for entry into the profession, the availability of a user-friendly assessment instrument with sufficient reliability and validity evidence provides a mechanism for fair and credible performance assessment across different placement settings and over time.

Reliable assessment depends, in part, on the use of psychometrically robust tools that provide clear and consistent scoring and interpretation guidelines for clinical supervisors. From a systematic review of the literature, the APP was rated equal or higher than all other international physiotherapy clinical performance assessment tools available [12]. The original APP

development work provided initial validity (e.g., dimensionality, differential item functioning, discriminant) and reliability (e.g., inter-rater) evidence [3–5]. The APP was designed to provide a unidimensional assessment of entry-level competency, whereby 20 items are aggregated into a single factor or total score representing overall minimal clinical competence to practice [3,4]. However, alternative interpretations suggest that overall physiotherapy entry-level competence could be better represented by two dimensions: professional behaviours (items 1–4) and clinical skills (items 5–20) [7]. The descriptive labelling of these dimensions as 'professional' and 'clinical' stems from thematic analysis of item content and indicators associated with each factor. Given this evidence, there is an ongoing question of whether entry-level competency is best captured as a global, single-factor or two-dimensional construct comprised of professional and clinical factors.

Generalisation of evidence is a cornerstone of the scientific process, serving as a litmus test for the validity and applicability of knowledge. There are numerous ways by which generalisation can be addressed, including but not limited to representative sampling, replication, cross-validation, and consideration of social-cultural and temporal considerations [29]. Individual studies, particularly those with small-to-moderately sized samples, that are imprecise representations of the population, often provide insufficient evidence to draw firm conclusions regarding the psychometric integrity of measurement instruments for science and practice [29]. Owing to its success in related areas of the human sciences [30], we took a 'Big Team' science approach to maximise inferences regarding the generalisation of evidence, whereby a large group of collaborators worked together to address a common goal [31]. In 2020, approximately 40 university staff and researchers across 23 ANZ universities established the Physiotherapy Clinical Education Research Collaborative (PCERC). This collaboration created a large and broad representative sample of entry-level physiotherapy placement activity across ANZ. In so doing, our representative sample of physiotherapy students undertaking real-world placements maximises generalisation because the findings are applicable to the broader population and breadth of placement environments, and reduces bias, increases external validity, facilitates meaningful comparisons, supports statistical inference, and enhances the overall credibility of the research. We complement this sampling approach to representativeness with a statistical framework that permits direct tests of the extent to which our findings generalise across people, time, and context.

In this paper, we report our inaugural project; a large, multi-site replication study, where we aimed to provide the physiotherapy profession with strong evidence regarding scoring protocols for assessment of entry-level performance using the APP. Our specific research questions were:

1. Do one or two factors best represent entry-level physiotherapy student performance scores obtained via the APP and what is the nature of their characterisation?

2. To what extent are the same number of factors and their interpretation, and item scaling captured equally over time and across contexts for assessments of entry-level physiotherapy student performance via the APP?

## Method

### Design

This multi-site study replicated the methodology of a single-site study [7], by using archival student APP assessment data from ANZ universities offering entry-level physiotherapy programs. We decided on course completion in 2019 as the target cohort because many clinical placements in 2020, and beyond, were disrupted by the COVID-19 pandemic.

## Ethics statement

Curtin University's Human Research Ethics Committee approved the study (HRE2021–0333) and provided approval for consent waiver. All participating universities gained reciprocal ethics and other governance approvals as required.

## Inclusivity in global research

Additional information regarding the ethical, cultural, and scientific considerations specific to inclusivity in global research is included in the Supporting Information (S1 Checklist).

## Participants

**Eligibility.** The lead author invited all ANZ universities offering entry-level physiotherapy programs in 2020 (n=25) to participate. Of these, 21 universities had graduating students in 2019 and utilised the APP to evaluate clinical placement performance. The remaining four were yet to have graduates (n=3) or didn't utilise the APP (n=1). Participants included entry-level physiotherapy students from eligible ANZ universities and programs who completed their final APP-assessed placement in 2019.

**Data collection.** Each eligible physiotherapy program collated data for their site in a standardised Microsoft Excel template. Site custodians extracted deidentified APP data scores from June 2022 to February 2023, representing the clinical supervisors' assessment of student performance upon completion of each placement in the participants' penultimate and/or final year, from paper and/or electronic records, and student demographic data from institution databases (see Table 1 for student and placement characteristics). DG managed the data collection and integration process using the secure CloudStor digital platform hosted by Australia's Academic and Research Network.

## Outcome measure

The APP is the primary physiotherapy assessment tool used to evaluate competency to practice in ANZ and has also been adopted in other countries [13,32]. The 20-item instrument covers seven key domains of physiotherapy practice, namely professional behaviour, communication, assessment, analysis and planning, intervention, evidence-based practice, and risk management (see S1 File) [3–5]. Clinical supervisors score each item, with reference to accompanying performance indicators, using a 5-point Likert scale (0 = infrequently/rarely demonstrates performance criteria to 4 = demonstrates most performance criteria to an excellent standard) to obtain a total score (out of 80), which is then represented as a percentage.

## Statistical analyses

We examined the research questions using raw item-level data via two independent yet related analytical phases using *Mplus* 8.10 [33]. Our analyses relied on full information maximum likelihood estimation with robust standard errors (MLR) to utilise all data to estimate models, with listwise deletion for analyses where missing data existed on personal or contextual factors. We calculated omega (ω) as an estimate of internal reliability for latent factors of our preferred structural representation [34]. All analysis scripts and outputs are available on the Open Science Framework (https://bit.ly/PCERC-app).

**Factorial validity evidence.** We examined the structural representation of APP scores via confirmatory factor analysis (CFA) and penalised structural equation models (PSEM) [35] with robust maximum likelihood estimation and cluster-robust standard errors to account for dependence among observations within each program. Both CFA and PSEM permit

**Table 1. Characteristics of students and placements.**

| Characteristic | Students (N = 1865) |
|---|---|
| **Gender, n (%)[a]** | |
| Male | 820 (44) |
| Female | 1045 (56) |
| **Age at 1st APP placement (y), mean (SD)** | 23.8 (4) |
| **Enrolled program, n (%)** | |
| Bachelor | 765 (41) |
| Bachelor with Honours | 309 (16.6) |
| Combined Bachelor and Masters | 140 (7.5) |
| Graduate Entry Masters | 434 (23.3) |
| Extended Masters/Doctor of Physiotherapy | 217 (11.6) |
| **Citizenship, n (%)** | |
| Australian | 881 (47.2) |
| International | 121 (6.5) |
| Unknown | 863 (46.3) |
| **Main Language, n (%)** | |
| English | 754 (40.4) |
| Non-English | 116 (6.2) |
| Unknown | 995 (53.4) |
| **Characteristic** | **Placements (N = 8979)** |
| **Placement Type, n (%)** | |
| Cardiorespiratory | 1800 (20) |
| Musculoskeletal | 2277 (25.4) |
| Neurological | 1898 (21.1) |
| Combination of 3 or more primary clinical areas | 1399 (15.6) |
| Other | 1174 (13.1) |
| Unknown | 431 (4.8) |
| **Placement Setting, n (%)[b]** | |
| Hospital - inpatients | 4699 (52.3) |
| Hospital - outpatients | 1072 (11.9) |
| Hospital - unknown | 643 (7.2) |
| Other (e.g., disability, residential aged care, community) | 2020 (22.5) |
| Combination[c] | 96 (1.1) |
| Unknown[c] | 449 (5) |
| **Duration of APP-assessed placements, n (%)** | |
| <5 weeks | 230 (2.6) |
| 5 weeks | 8289 (92.3) |
| >5 weeks | 460 (5.1) |
| **Location of APP-assessed placements in course, n (%)** | |
| Penultimate year | 2384 (26.6) |
| Final year | 6595 (73.4) |
| **Placement Outcome, n (%)** | |
| Pass | 8770 (97.7) |
| Fail | 209 (2.3) |

[a] *Data often captured on entry to university programs; We acknowledge that students might identify in ways that differ from these two categories.*

[b] *There existed numerous and diverse ways by which programs classified placement settings; we manually categorised them into meaningful setting areas for the purposes of examining measurement invariance across placement contexts.*

[c] *Excluded from measurement invariance analyses due to small sample size for these groups.*

investigations of structural representations by modelling associations between observed scores (raw item-level data) and latent factors (i.e., professional or clinical) hypothesised to explain patterns in responses, with unexplained or 'leftover' variance captured in residual variances. The main difference between CFA and PSEM is the precision with which latent variables and observed scores are structurally linked. As depicted in Fig 1, the strict hypothesis in CFA is that item indicators reflect one latent factor only, whereas item indicators can load on their intended factor and cross-load on other latent factors in PSEM [36]. In PSEM, priors are employed to capture assumptions about certain parameters that act like guidelines or weights, influencing the analysis by nudging it towards these initial expectations [35]. However, as actual data comes in, these priors are updated based on how well they match the data. This process helps in managing complex models, especially when dealing with many variables or when the data is sparse or uncertain. Essentially, priors in PSEM blend what you initially think (your prior beliefs) with what the data is telling you, leading to robust conclusions. In all PSEM models, we used the alignment loss function prior (0,1) to approximate zero for item cross-loadings on unintended factors.

We examined 13 possible structural representations across three broad categories: a unidimensional model in which all 20 items are explained by a single latent factor, as per the original protocol [3]; correlated two-factor solutions in which latent factors characterise 'professional' and 'clinical' domains [6–8]; and bifactor solutions in which a global factor – entry-level competency – occurs alongside specific factors of professional and/or clinical domains which

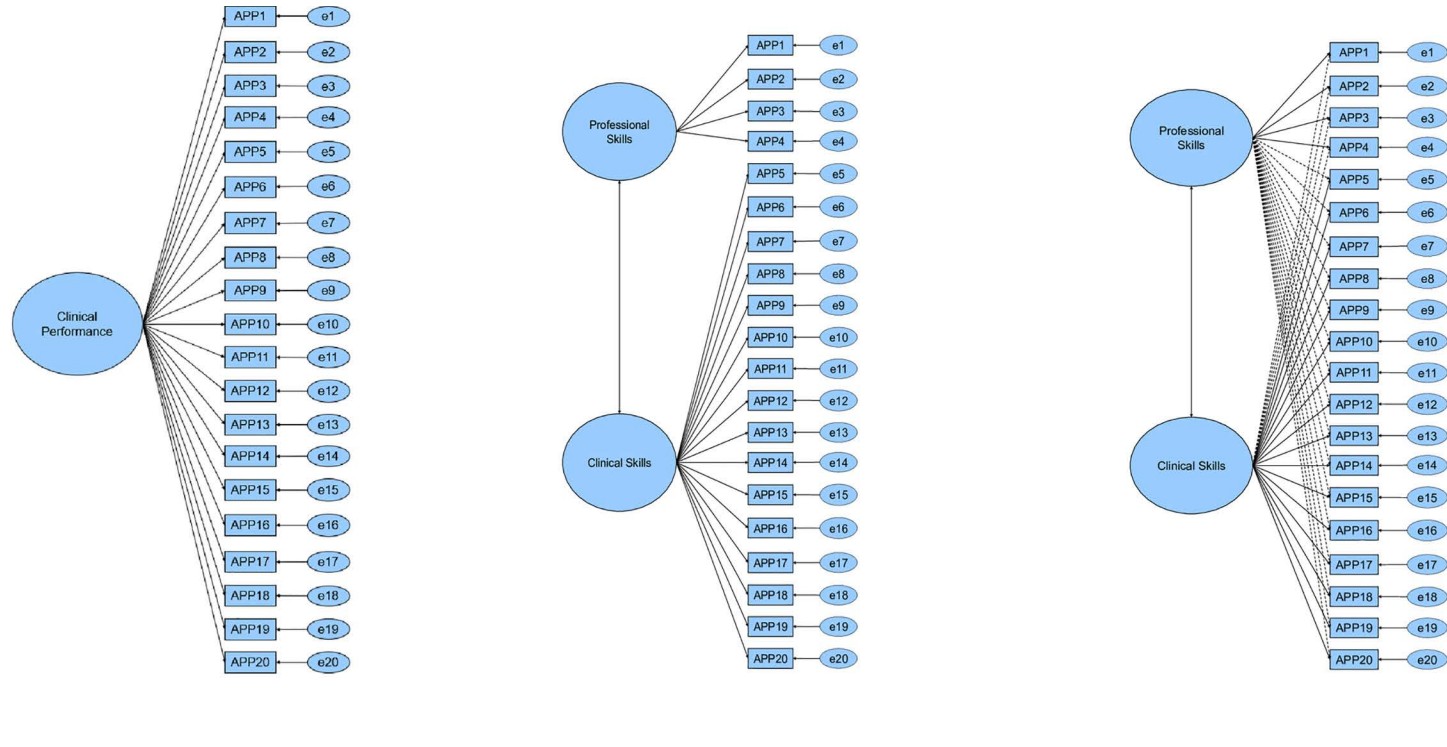

**One-factor CFA**              **Two-factor CFA**              **Two-factor PSEM**

**Fig 1. Visual depiction of one-factor and two-factor models for confirmatory factor analysis (CFA) and penalised structural equation modelling (PSEM).** Elipses (circles) represent latent variables, boxes represent observed variables, single-headed arrows represent a directional effect of a latent variable on an observed variable, double-headed arrows represent correlation among latent factors, solid lines represent target factor loading and dashed lines represent non-target factor loading. APP = Assessment of Physiotherapy Practice item, e = residual variance.

are anchored to a subset of the overall item pool. We assessed the degree of model-data fit via a multifaceted approach, relying primarily on the comparative fit index (CFI), Tucker-Lewis index (TLI), and the root mean square error of approximation (RMSEA), with values of ≥.90 for CFI and TLI and ≤.08 for RMSEA considered to reflect acceptable fit [37]. Regarding model selection, we prioritised model superiority for lower values for the Akaike Information Criteria (AIC), Bayesian Information Criteria (BIC), and sample sized adjusted BIC (ABIC) [38]; and sound intended factor loadings (~ >.40) and small cross-loadings (~ <.20). In so doing, we preferred an approach where meaningful intended factor loadings can be differentiated quantitatively from meaningful cross-loadings, rather than rely on the rule of thumb of ≥.32 for meaningful factor loadings [39]. Said differently, we prioritised an inference framework where the difference in magnitude of loadings for each item on their intended versus unintended factor were approximately .20. Although we monitored model-data fit indices to evaluate the overall model quality, we focused primarily on conceptually informed item retention because purely statistical benchmarks can undermine construct representation (e.g., breadth, depth). Thus, when an item with a statistically borderline loading represented a theoretically essential aspect of the target construct, we prioritised content validity over strict adherence to cutoff values.

**Measurement invariance evidence.** Meaningful group comparisons using aggregate test scores rest on the assumption that latent variables are captured via the same origin and scale and therefore share the same operational definition and meaning across (sub)populations [40,41]. Achieving measurement invariance is crucial for comparing scores between groups or over time, as it ensures that observed differences reflect true variations in the construct rather than inconsistencies in measurement. The subpopulations of interest within our sample include repeated assessments and three elements that characterise contextual features of the professional practice landscape within ANZ, namely placement type (cardiorespiratory, musculoskeletal, neurological, other, or some combination of them), setting (hospital-inpatient, hospital-outpatient, hospital-unknown, or other), and the year in which placements occurred (final year only or penultimate and final years). Measurement invariance evidence is essential for inferences regarding the generalisation of concepts and assessment properties. Evidence that supports measurement invariance across time and context provides confidence that assessment instruments function roughly equivalently irrespective of the circumstances in which they are applied.

Testing measurement invariance involves sequentially comparing nested models in which there are increasingly stricter equality constraints. We deployed the widely accepted 3-stage approach to estimate if the (1) number of factors and items per factor (configural), (2) magnitude of factor loadings (metric), and (3) magnitude of factor loadings and item intercepts (scalar) are equivalent over time and across contextual factors. For nested model comparisons, we relied on changes in fit indices rather than $\mathcal{X}^2$ differences because they are insensitive to sample size and minor misspecifications [42]. Regarding changes in model-data fit indices, we considered a decline of CFI and TLI of 0.01 or less, and an increase in RMSEA of 0.015 or less to support parsimony or invariance between nested models [43].

## Results

### Flow of participants

In total, 19 of 21 eligible universities (90.5% response rate) obtained the necessary ethics and governance approvals and provided APP data from a minimum of two, APP-assessed placements per student (see Table 2 for overview). Individual data included 1865 students who collectively completed 9387 APP-assessed placements. We retrieved complete APP records for 8979 (95.6%) placements, providing comprehensive coverage of entry-level physiotherapy placements for the graduating students of 2019 across ANZ (see Table 1 for overview).

**Table 2. Characteristics of participating university programs.**

| Characteristic | |
|---|---|
| **Universities, n** | 19 |
| **Locations where entry-level programs offered, n[a]** | 21 |
| **Australia (n=20)** | |
| New South Wales/Australian Capital Territory | 7 |
| Queensland | 6 |
| South Australia | 1 |
| Victoria | 4 |
| Western Australia | 2 |
| **New Zealand (n=1)** | 1 |
| **Programs offered (n=36), n (%)** | |
| Bachelor | 10 (28) |
| Bachelor with Honours | 12 (33) |
| Combined Bachelor and Masters | 2 (6) |
| Graduate Entry Masters | 9 (25) |
| Extended Masters/Doctor of Physiotherapy | 3 (8) |
| **Number of APP-assessed placements in programs offered (mean = 5)[b]** | |
| Three | 1 |
| Four | 10 |
| Five | 15 |
| Six | 9 |
| Seven | 1 |
| **Duration of APP-assessed placements (number of programs), n[c]** | |
| <5 weeks | 2 |
| 5 weeks | 34 |
| >5 weeks | 3 |
| **Location of APP-assessed placements within course, n (%)** | |
| Both penultimate and final years | 22 (61) |
| Final year only | 14 (39) |

[a]*Two universities conduct programs at two different locations; these are counted separately because they require staffing and resources at multiple sites*

[b]*A proportion of students from one university (40 out of 46) undertook seven placements; these data are included in the table but were excluded from data analyses due to the small number/sample*

[c]*Some programs offer placements of varying durations and are captured in more than one category*

## Factorial validity evidence

Model fit statistics for the total sample and each placement individually are presented in S2 File. Information criteria generally supported the superiority of the PSEM bifactor model with four or six items for the professional dimension relative to the other models tested. All models evidenced acceptable model-data fit according to CFI, TLI, and RMSEA values, except for the PSEM bifactor models with four or six items for professional. The next best working models are the PSEM 2-factor (four or six items for professional dimension) and PSEM bifactor (specific factor anchored to four or six items for professional dimension) representations. However, all models except for the PSEM 2-factor (four items for professional) evidenced an inadequate profile of factors as represented by intended factor loadings and cross-loadings (S2 File). As recommended by one reviewer, we compared and contrasted findings between MLR and weighted least square mean and variance adjusted (WLSMV) estimation. Model-data fit

and factor loading estimates obtained with WLSMV estimation are provided in the Supporting Information (S1 and S3 Tables in S2 File). Overall, all models evidenced acceptable model-data fit according to CFI, TLI, and RMSEA values, except for the PSEM bifactor (4 items for professional) representation which did not converge as an unidentified model. Regarding our preferred PSEM 2-factor (4 items for professional and 16 items for clinical) solution, differences in factor loadings were minimal (0–0.15) and consistent with interpretations of intended factor loadings and cross-loading benchmarks between MLR and WLSMV estimation.

Collectively, therefore, model selection indices and factor loadings suggest the 2-factor PSEM model with four items for the professional dimension and 16 items for the clinical dimension is the best approximation of reality regarding these APP data. Factor loadings for this model are presented in Table 3. Most APP items loaded meaningfully onto their respective latent factor (>.40), with higher and more consistent loadings observed for the clinical dimension, and few substantial cross-loadings (e.g., items 2 and 20). One reviewer suggested we remove these poorly fitting items according to statistical criteria alone then recalculate the factor model, and do so iteratively until a well-defined model is achieved. In essence, iteratively modifying the model based on observed loadings within the same dataset can capitalise on chance variation, leading to overfitting and compromised generalisability. This approach resembles *post hoc* model specification, where decisions about item retention and factor structure are made after examining the data rather than being guided by a priori theoretical or empirical criteria. The iterative removal of poorly fitting items based on statistical criteria alone, followed by re-estimation of the model with the same sample, can artificially inflate model fit and lead to a final structure that may poorly generalise to other samples. Most

**Table 3. Standardised factor loadings and latent variable correlation of the two-factor measurement model for penalised structural equation model with the total sample (APP = Assessment of Physiotherapy Practice item; grey shade = statistically significant loading).**

| Factor | Item | β | p | Factor | Item | β | p |
|---|---|---|---|---|---|---|---|
| Professional | APP1 | 0.829 | <.001 | Clinical | APP1 | 0.021 | 0.068 |
| Professional | APP2 | 0.468 | <.001 | Clinical | APP2 | 0.315 | <.001 |
| Professional | APP3 | 0.868 | <.001 | Clinical | APP3 | -0.005 | 0.515 |
| Professional | APP4 | 0.375 | <.001 | Clinical | APP4 | 0.415 | <.001 |
| Professional | APP5 | 0.236 | <.001 | Clinical | APP5 | 0.581 | <.001 |
| Professional | APP6 | 0.161 | <.001 | Clinical | APP6 | 0.572 | <.001 |
| Professional | APP7 | 0.142 | <.001 | Clinical | APP7 | 0.66 | <.001 |
| Professional | APP8 | -0.021 | 0.167 | Clinical | APP8 | 0.813 | <.001 |
| Professional | APP9 | 0.016 | 0.436 | Clinical | APP9 | 0.791 | <.001 |
| Professional | APP10 | -0.021 | 0.214 | Clinical | APP10 | 0.835 | <.001 |
| Professional | APP11 | -0.031 | 0.075 | Clinical | APP11 | 0.847 | <.001 |
| Professional | APP12 | 0.003 | 0.841 | Clinical | APP12 | 0.762 | <.001 |
| Professional | APP13 | -0.027 | 0.061 | Clinical | APP13 | 0.857 | <.001 |
| Professional | APP14 | 0.049 | 0.01 | Clinical | APP14 | 0.781 | <.001 |
| Professional | APP15 | 0.055 | 0.015 | Clinical | APP15 | 0.73 | <.001 |
| Professional | APP16 | 0.042 | 0.031 | Clinical | APP16 | 0.849 | <.001 |
| Professional | APP17 | -0.088 | <.001 | Clinical | APP17 | 0.88 | <.001 |
| Professional | APP18 | 0.011 | 0.555 | Clinical | APP18 | 0.75 | <.001 |
| Professional | APP19 | 0.129 | <.001 | Clinical | APP19 | 0.588 | <.001 |
| Professional | APP20 | 0.313 | <.001 | Clinical | APP20 | 0.509 | <.001 |
| Clinical WITH Professional | | 0.708 | 0 | | | | |

importantly, ignoring conceptual considerations for the removal of items would inevitably weaken our confidence in the content validity of the Assessment of Physiotherapy Practice (APP). Removing items with uncertain statistical properties in our analyses – "demonstrates collaborative practice", "commitment to learning", "verbal and non-verbal communication", and "identifies adverse events/near misses and minimises risk associated with assessment and interventions" – would effectively remove essential content that is required for entry-level physiotherapy performance. The latent factor correlation between the professional (ω =.80) and clinical (ω =.96) dimensions was moderately strong.

## Measurement invariance evidence

Model fit statistics for the invariance tests for the PSEM 2-factor representation with four items for professional and 1-factor CFA model are presented in Table 4. We chose to examine and present the findings of the 1-factor model because it is the original operationalisation of APP for student performance [3–5] and commonly implemented among clinical education programs in ANZ. Overall, model-data fit statistics supported scalar invariance for both the 1-factor and 2-factor solutions across time and contextual factors. Convergence issues – likely because of the limited sample size for placements less than 5 weeks (n = 230) or more than 5 weeks (n = 460) in duration relative to those which were around 5 weeks in duration (n = 8289) – meant that we were unable to test measurement invariance of the APP across placement length. Factor analyses (PSEM) supported the structural validity of the 2-factor model with data obtained from placements which were around 5 weeks in duration; currently, there is an absence of evidence for the structural validity for placements which are less or more than 5 weeks in duration.

## Discussion

Our multi-site replication study supported the psychometric integrity of the 1-factor [3–5] and 2-factor [7] representations of entry-level physiotherapy performance utilising the APP. Relatively speaking, model comparison data support the superiority of the 2-factor model with four items within the professional dimension and 16 items within the clinical dimension. This representation remained largely consistent over time and across diverse practice areas in both penultimate and final years of study.

Entry-level physiotherapy performance is a complex, multidimensional concept. Typically, students must demonstrate diverse skills spanning, at a minimum, cognitive (e.g., clinical reasoning), technical/physical (e.g., manual therapy techniques), and interpersonal (e.g., communication with supervisors and patients) components of practice [14,20]. Accurate assessment relies, in part, on the availability of psychometrically sound assessment instruments, with consistent scoring and interpretation guidelines, for clinical supervisors to use [44]. Drawing from a large representative sample of student performances, we provide strong evidence regarding the internal psychometric properties of the APP including measurement validity and reliability which generalises across the ANZ physiotherapy entry-level placement context as well as temporal and contextual considerations that characterise the complexities of real-world settings. We showed that interpretation of entry-level physiotherapy performance is best operationalised via the APP as a 2-factor concept with domains characterised by professional (items 1–4) and clinical (items 5–20) indicators. The 2-factor representation aligns with contemporary practice, whereby healthcare graduates should be professional, ethically, and legally responsible as well as technically-skilled. Delivering physiotherapy care via assessment, analysis, and planning together with high-quality professional behaviours such as patient rights and consent is essential for safe and effective practice, as per contemporary

**Table 4. Summary of fit indices for measurement invariance analyses.**

| Model | AIC | BIC | ABIC | $X2$ | df | $p$ | CFI | TLI | RMSEA | RMSEA 90% CI | | ΔCFI | ΔTLI | ΔRMSEA |
|---|---|---|---|---|---|---|---|---|---|---|---|---|---|---|
| CFA 1-factor_placement number (configural) | 261202.41 | 26355.21 | 262611.19 | 9373.65 | 1020 | <.001 | 0.947 | 0.941 | 0.074 | 0.073 | 0.076 | | | |
| CFA 1-factor_placement number (metric) | 261120.53 | 262999.68 | 262157.55 | 9600.98 | 1115 | <.001 | 0.946 | 0.945 | 0.072 | 0.070 | 0.073 | -0.001 | 0.004 | -0.002 |
| CFA 1-factor_placement number (scalar) | 261188.94 | 262394.43 | 261854.20 | 9822.98 | 1210 | <.001 | 0.945 | 0.948 | 0.069 | 0.068 | 0.071 | -0.001 | 0.003 | -0.003 |
| CFA 1-factor_placement multiple years (configural) | 264491.07 | 265343.38 | 264962.04 | 8492.55 | 340 | <.001 | 0.940 | 0.933 | 0.073 | 0.072 | 0.074 | | | |
| CFA 1-factor_placement multiple years (metric) | 264518.46 | 265235.83 | 264914.87 | 8159.70 | 359 | <.001 | 0.943 | 0.939 | 0.070 | 0.068 | 0.071 | 0.003 | 0.006 | -0.003 |
| CFA 1-factor_placement multiple years (scalar) | 264645.99 | 265228.41 | 264967.83 | 7412.16 | 378 | <.001 | 0.948 | 0.948 | 0.064 | 0.063 | 0.064 | 0.005 | 0.009 | -0.006 |
| CFA 1-factor_placement type (configural) | 251339.25 | 252455.29 | 252501.94 | 8046.35 | 850 | <.001 | 0.947 | 0.941 | 0.070 | 0.069 | 0.072 | | | |
| CFA 1-factor_placement type (metric) | 251351.96 | 252931.93 | 252220.10 | 8177.49 | 926 | <.001 | 0.947 | 0.945 | 0.068 | 0.066 | 0.069 | 0.000 | 0.004 | -0.002 |
| CFA 1-factor_placement type (scalar) | 252171.76 | 253215.68 | 252745.36 | 8508.20 | 1002 | <.001 | 0.945 | 0.948 | 0.066 | 0.065 | 0.067 | -0.002 | 0.003 | -0.002 |
| CFA 1-factor_placement setting (configural) | 247801.11 | 247490.72 | 247728.04 | 8834.93 | 680 | <.001 | 0.964 | 0.960 | 0.075 | 0.074 | 0.077 | | | |
| CFA 1-factor_placement setting (metric) | 247844.12 | 249132.45 | 248550.91 | 8738.51 | 737 | <.001 | 0.965 | 0.964 | 0.072 | 0.070 | 0.073 | 0.001 | 0.004 | -0.003 |
| CFA 1-factor_placement setting (scalar) | 248668.65 | 249555.69 | 249155.29 | 8705.15 | 794 | <.001 | 0.965 | 0.967 | 0.069 | 0.067 | 0.070 | 0.000 | 0.003 | -0.003 |
| PSEM 2-factor_placement number (configural) | 255176.09 | 257913.26 | 256686.62 | 6579.16 | 994 | <.001 | 0.964 | 0.959 | 0.062 | 0.060 | 0.063 | | | |
| PSEM 2-factor_placement number (metric) | 255090.42 | 257118.48 | 256209.62 | 5849.92 | 1094 | <.001 | 0.970 | 0.968 | 0.054 | 0.053 | 0.056 | 0.006 | 0.009 | -0.008 |
| PSEM 2-factor_placement number (scalar) | 255121.28 | 256511.14 | 255888.28 | 5509.39 | 1184 | <.001 | 0.972 | 0.974 | 0.050 | 0.048 | 0.051 | 0.002 | 0.006 | -0.004 |
| PSEM 2-factor_placement multiple years (configural) | 104194.61 | 105072.93 | 104621.72 | 2789.84 | 318 | <.001 | 0.966 | 0.960 | 0.066 | 0.064 | 0.068 | | | |
| PSEM 2-factor_placement multiple years (metric) | 104176.63 | 104931.24 | 104543.59 | 2611.18 | 338 | <.001 | 0.969 | 0.965 | 0.061 | 0.059 | 0.063 | 0.003 | 0.005 | -0.005 |
| PSEM 2-factor_placement multiple years (scalar) | 104195.66 | 104838.94 | 104508.48 | 2478.31 | 356 | <.001 | 0.971 | 0.969 | 0.058 | 0.056 | 0.060 | 0.002 | 0.004 | -0.003 |
| PSEM 2-factor_placement type (configural) | 245590.85 | 247883.22 | 246850.43 | 5571.75 | 825 | <.001 | 0.950 | 0.960 | 0.058 | 0.057 | 0.059 | | | |
| PSEM 2-factor_placement type (metric) | 245674.05 | 247402.15 | 246623.58 | 5010.24 | 905 | <.001 | 0.970 | 0.968 | 0.052 | 0.050 | 0.053 | 0.020 | 0.008 | -0.006 |
| PSEM 2-factor_placement type (scalar) | 246371.02 | 247591.27 | 247041.51 | 5099.97 | 977 | <.001 | 0.970 | 0.971 | 0.050 | 0.048 | 0.051 | 0.000 | 0.003 | -0.002 |
| PSEM 2-factor_placement length (configural) | 258440.36 | 259882.20 | 259237.10 | 28624.79 | 487 | <.001 | 0.793 | 0.758 | 0.139 | 0.138 | 0.140 | | | |
| PSEM 2-factor_placement length (metric) | 258421.86 | 259579.59 | 259061.60 | 19341.26 | 527 | <.001 | 0.862 | 0.851 | 0.109 | 0.108 | 0.111 | 0.069 | 0.093 | -0.030 |
| PSEM 2-factor_placement length (scalar) | 258529.59 | 259431.62 | 259028.04 | 16565.52 | 563 | <.001 | 0.882 | 0.881 | 0.097 | 0.096 | 0.099 | 0.020 | 0.030 | -0.012 |
| PSEM 2-factor_placement setting (configural) | 242134.49 | 243993.06 | 243154.12 | 5891.66 | 656 | <.001 | 0.977 | 0.973 | 0.062 | 0.060 | 0.063 | | | |
| PSEM 2-factor_placement setting (metric) | 242218.05 | 243654.22 | 243005.94 | 5155.54 | 716 | <.001 | 0.980 | 0.979 | 0.054 | 0.053 | 0.056 | 0.003 | 0.006 | -0.008 |
| PSEM 2-factor_placement setting (scalar) | 242981.71 | 244037.71 | 243561.04 | 5164.16 | 770 | <.001 | 0.981 | 0.981 | 0.052 | 0.051 | 0.053 | 0.001 | 0.002 | -0.002 |

standards for entry-level performance and registration requirements [15,17]. The 2-factor representation also enhances the utility of the APP for educators and supervisors by highlighting practice areas where students may need focussed support or development. In summary, to meet entry-level standards, physiotherapy students should be assessed as competent across both professional and clinical dimensions of physiotherapy practice.

Despite the firm conclusion regarding the dimensionality of entry-level physiotherapy performance, there are important nuances to the data which require consideration for future use. On the surface, the PSEM 2-factor (four items for professional) model provided the best representation of entry-level competence, yet some item level data were less optimal. Specifically, item 4 "demonstrates collaborative practice" had almost identical factor loadings across professional and clinical dimensions (0.375 vs 0.415 respectively), whereas three items (2 – commitment to learning, 5 – verbal and non-verbal communication, and 20 – identifies adverse events/near misses and minimises risk associated with assessment and interventions) loaded higher on their intended factor relative to the unintended factor, but failed to meet one or both criteria (>.40 on intended and <.20 on unintended factor). Performance indicators within item 4 include "works collaboratively & respectfully with support staff" and "collaborates with the health care team & client to achieve optimal outcomes". At face value, these features likely cut across most, if not all aspects, of entry-level physiotherapy performance. Commitment to learning (item 2) includes some performance indicators that are subjective in nature, such as "takes responsibility for learning…" and "demonstrates self-evaluation…". Items that require less subjectivity exhibited higher factor loadings, emphasising the need for clarity in APP item indicators, especially considering the dynamic nature of contemporary healthcare settings [20]. Relying solely on statistical criteria to make inferences regarding item selection and retainment would likely mean that we'd need to compromise the content validity of entry-level physiotherapy performance in some way. Rather than throw the baby out with the bathwater, future work is required to refine these ambiguous items to maximise their conceptual clarity and scoring precision. The conceptual feature of 'demonstrates collaborative practice', for example, could be partitioned into separate items that specify with precision exactly what this collaborative approach looks like for professional and clinical elements of performance. Item enhancements could also align with item content in the 2023 update in physiotherapy practice standards within ANZ [17]. As reliability and validity are properties of test scores rather than instruments themselves, we advocate for ongoing validation work on the current version of the APP or any item refinements to the conceptual space.

## Strengths and limitations

Key strengths of this study include the utilisation of a Big Team science approach with a large, heterogenous, representative sample and minimal missing data points, alongside rigorous statistical analyses. This combination improves efficiency, precision, confidence, and generalisability of study findings within the ANZ entry-level physiotherapy context [29]. Our study addressed one of the key limitations of existing evidence [4,5,7] by supporting the 2-factor representation and item scaling across diverse geographic locations, placement sites and settings, and supervisor demographics. We acknowledge that the availability of a psychometrically supported instrument for assessment of professional competency is only one piece of the puzzle for maximising robust inferences regarding individual performance [24]. Assessment and learning within complex workplaces, requires an integrative and holistic approach that considers performance within a socio-cultural context, where social interactions, and human judgement and bias influence performance and decision-making [26]. This holistic approach underscores the importance of assessors having sufficient understanding of the physiotherapy

practice content and assessment literacy, such as expected standards and behaviours, and how to interpret observations [24,26,45]. It also requires a shift away from the assessment instrument and a focus onto supporting and improving human judgement. Finally, our analyses focused on the internal psychometric properties of the APP. Future research is required to gather knowledge on external validity evidence, particularly predictive validity.

## Conclusion

This large multi-site study provides the physiotherapy profession with the necessary evidence to move towards a standardised application of the ANZ-adopted entry-level physiotherapy performance assessment instrument. Our data support the superiority of the 2-factor model with four items for the professional and 16 items for the clinical dimension, yet the original 1-factor is also a viable representation of the APP. Consistency in professional competency assessment will permit improved benchmarking and quality assurances for accreditation and professional registration requirements and, ultimately, high-quality educational models for training and assessing the future generation of physiotherapy professionals. These findings are also important as others adopt or adapt the APP for assessments of entry-level physiotherapy performance globally [13,32,46].

## Supporting information

**S1 Checklist.  Inclusivity in global research questionnaire.**
(DOCX)

**S1 File.  The 20-item Assessment of Physiotherapy Practice and examples of performance indicators.**
(PDF)

**S2 File.  Supporting results information for model-data fit statistics and factor loadings.**
(XLSX)

## Acknowledgements

The authors thank numerous individuals at each university and placement site who supported this project including but not limited to clinical supervisors who completed APP assessments, as well as academic and professional staff who contributed to the operational activities (e.g., data collection, governance).
Membership of the Physiotherapy Clinical Education Research Collaborative includes:

1.  Alison Bell, Allied Health and Human Performance, University of South Australia, Adelaide, Australia

2.  Allyson Calder, Centre for Health, Activity and Rehabilitation Research, School of Physiotherapy, University of Otago, Dunedin, New Zealand

3.  Julie Gauchwin, School of Allied Health, Australian Catholic University, Brisbane, Australia

4.  Chris Higgs, School of Physiotherapy, University of Otago, Dunedin, New Zealand

5.  Leanne Johnston, School of Health and Rehabilitation Sciences, The University of Queensland, Brisbane, Australia

6.  Chantal Maher, Graduate School of Health, University of Technology Sydney, Sydney, Australia

7. Nikki Milne, Department of Physiotherapy, Faculty of Health Sciences and Medicine, Bond University, Gold Coast, Australia

8. Tim Newing, School of Health Sciences, The University of Notre Dame Australia, Fremantle, Australia

9. Gitte Nielsen, College of Healthcare Sciences, James Cook University, Townsville, Australia

10. Emma Richards, Graduate School of Health, University of Technology Sydney, Sydney, Australia

11. Gisela Van Kessel, UniSA Online, University of South Australia, Adelaide, Australia

12. Jill Williams, College of Nursing and Health Sciences, Flinders University, Adelaide, Australia

## Author contributions

**Conceptualization:** Alan Reubenson, Leo Ng, Vidya Lawton, Irmina Nahon, Rebecca Terry, Rosemary Corrigan, Megan Dalton, Amabile Borges Dario, Michael Donovan, Ruth Dunwoodie, Genevieve M. Dwyer, Roma Forbes, Alison Francis-Cracknell, Andrea Hams, Taryn Jones, Tanya Palmer, Casey Peiris, Carolyn Taylor, Debra Virtue, Daniel F. Gucciardi.

**Data curation:** Alan Reubenson, Vidya Lawton, Irmina Nahon, Rebecca Terry, Claire Baldwin, Julia Blackford, Alex Bond, Rosemary Corrigan, Megan Dalton, Amabile Borges Dario, Michael Donovan, Ruth Dunwoodie, Genevieve M. Dwyer, Roma Forbes, Janelle Gill, Andrea Hams, Anne Jones, Taryn Jones, Belinda Judd, Ewan Kennedy, Prue Morgan, Tanya Palmer, Casey Peiris, Carolyn Taylor, Debra Virtue, Cherie Zischke, Daniel F. Gucciardi.

**Formal analysis:** Alan Reubenson, Daniel F. Gucciardi.

**Investigation:** Alan Reubenson, Vidya Lawton, Irmina Nahon, Rebecca Terry, Claire Baldwin, Julia Blackford, Alex Bond, Rosemary Corrigan, Amabile Borges Dario, Michael Donovan, Genevieve M. Dwyer, Roma Forbes, Janelle Gill, Anne Jones, Taryn Jones, Belinda Judd, Ewan Kennedy, Tanya Palmer, Casey Peiris, Carolyn Taylor, Debra Virtue, Daniel F. Gucciardi.

**Methodology:** Alan Reubenson, Leo Ng, Vidya Lawton, Irmina Nahon, Rebecca Terry, Daniel F. Gucciardi.

**Project administration:** Alan Reubenson, Leo Ng, Vidya Lawton, Irmina Nahon, Rebecca Terry, Claire Baldwin, Julia Blackford, Alex Bond, Rosemary Corrigan, Megan Dalton, Amabile Borges Dario, Michael Donovan, Ruth Dunwoodie, Genevieve M. Dwyer, Roma Forbes, Alison Francis-Cracknell, Janelle Gill, Andrea Hams, Anne Jones, Taryn Jones, Belinda Judd, Ewan Kennedy, Prue Morgan, Tanya Palmer, Casey Peiris, Carolyn Taylor, Debra Virtue, Cherie Zischke, Daniel F. Gucciardi.

**Resources:** Alan Reubenson, Leo Ng, Vidya Lawton, Irmina Nahon, Rebecca Terry, Claire Baldwin, Julia Blackford, Alex Bond, Rosemary Corrigan, Megan Dalton, Amabile Borges Dario, Michael Donovan, Ruth Dunwoodie, Genevieve M. Dwyer, Roma Forbes, Alison Francis-Cracknell, Janelle Gill, Andrea Hams, Anne Jones, Taryn Jones, Belinda Judd, Ewan Kennedy, Prue Morgan, Tanya Palmer, Casey Peiris, Carolyn Taylor, Debra Virtue, Cherie Zischke.

**Supervision:** Alan Reubenson, Leo Ng, Vidya Lawton, Irmina Nahon, Rebecca Terry, Daniel F. Gucciardi.

**Writing – original draft:** Alan Reubenson, Daniel F. Gucciardi.

**Writing – review & editing:** Alan Reubenson, Leo Ng, Vidya Lawton, Irmina Nahon, Rebecca Terry, Claire Baldwin, Julia Blackford, Alex Bond, Rosemary Corrigan, Megan Dalton, Amabile Borges Dario, Michael Donovan, Ruth Dunwoodie, Genevieve M. Dwyer, Roma Forbes, Alison Francis-Cracknell, Janelle Gill, Andrea Hams, Anne Jones, Taryn Jones, Belinda Judd, Ewan Kennedy, Prue Morgan, Tanya Palmer, Casey Peiris, Carolyn Taylor, Debra Virtue, Cherie Zischke, Daniel F. Gucciardi.

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
