## [Decision Letter · Decision Letter 0]

28 Oct 2024

PONE-D-24-38498The Assessment of Physiotherapy Practice is a robust measure of entry-level physiotherapy standards: Reliability and validity evidence from a large, representative samplePLOS ONE

Dear Dr. Reubenson,

Thank you for submitting your manuscript to PLOS ONE. After careful consideration, we feel that it has merit but does not fully meet PLOS ONE’s publication criteria as it currently stands. Therefore, we invite you to submit a revised version of the manuscript that addresses the points raised during the review process.

We look forward to receiving your revised manuscript.

Kind regards,

Henri Tilga, PhD

Academic Editor

PLOS ONE

Journal requirements: When submitting your revision, we need you to address these additional requirements. 1. Please ensure that your manuscript meets PLOS ONE's style requirements, including those for file naming. The PLOS ONE style templates can be found at https://journals.plos.org/plosone/s/file?id=wjVg/PLOSOne_formatting_sample_main_body.pdf and https://journals.plos.org/plosone/s/file?id=ba62/PLOSOne_formatting_sample_title_authors_affiliations.pdf 2. Please include a complete copy of PLOS’ questionnaire on inclusivity in global research in your revised manuscript. Our policy for research in this area aims to improve transparency in the reporting of research performed outside of researchers’ own country or community. The policy applies to researchers who have travelled to a different country to conduct research, research with Indigenous populations or their lands, and research on cultural artefacts. The questionnaire can also be requested at the journal’s discretion for any other submissions, even if these conditions are not met.  Please find more information on the policy and a link to download a blank copy of the questionnaire here: https://journals.plos.org/plosone/s/best-practices-in-research-reporting. Please upload a completed version of your questionnaire as Supporting Information when you resubmit your manuscript. 3. We note that you have indicated that there are restrictions to data sharing for this study. For studies involving human research participant data or other sensitive data, we encourage authors to share de-identified or anonymized data. However, when data cannot be publicly shared for ethical reasons, we allow authors to make their data sets available upon request. For information on unacceptable data access restrictions, please see http://journals.plos.org/plosone/s/data-availability#loc-unacceptable-data-access-restrictions.  Before we proceed with your manuscript, please address the following prompts: a) If there are ethical or legal restrictions on sharing a de-identified data set, please explain them in detail (e.g., data contain potentially identifying or sensitive patient information, data are owned by a third-party organization, etc.) and who has imposed them (e.g., a Research Ethics Committee or Institutional Review Board, etc.). Please also provide contact information for a data access committee, ethics committee, or other institutional body to which data requests may be sent. b) If there are no restrictions, please upload the minimal anonymized data set necessary to replicate your study findings to a stable, public repository and provide us with the relevant URLs, DOIs, or accession numbers. Please see http://www.bmj.com/content/340/bmj.c181.long for guidelines on how to de-identify and prepare clinical data for publication. For a list of recommended repositories, please see https://journals.plos.org/plosone/s/recommended-repositories. You also have the option of uploading the data as Supporting Information files, but we would recommend depositing data directly to a data repository if possible. Please update your Data Availability statement in the submission form accordingly. 4. One of the noted authors is a group [Physiotherapy Clinical Education Research Collaborative (PCERC)]. In addition to naming the author group, please list the individual authors and affiliations within this group in the acknowledgments section of your manuscript. Please also indicate clearly a lead author for this group along with a contact email address. 5. Your ethics statement should only appear in the Methods section of your manuscript. If your ethics statement is written in any section besides the Methods, please move it to the Methods section and delete it from any other section. Please ensure that your ethics statement is included in your manuscript, as the ethics statement entered into the online submission form will not be published alongside your manuscript.  6. Please include captions for your Supporting Information files at the end of your manuscript, and update any in-text citations to match accordingly. Please see our Supporting Information guidelines for more information: http://journals.plos.org/plosone/s/supporting-information. 

Reviewers' comments:

Reviewer's Responses to Questions

**Comments to the Author**

1. Is the manuscript technically sound, and do the data support the conclusions?

Reviewer #1: Partly

2. Has the statistical analysis been performed appropriately and rigorously? 

Reviewer #1: I Don't Know

3. Have the authors made all data underlying the findings in their manuscript fully available?

Reviewer #1: Yes

4. Is the manuscript presented in an intelligible fashion and written in standard English?

Reviewer #1: Yes

5. Review Comments to the Author

Reviewer #1: General comment:

Great paper of significant value to the physiotherapy profession. I think to make this more accessible to student supervisors and physiotherapy academics the language needs to be simplified, or complex terms given context. The other key factors that I believe needs to be explained further is why/how modelling the assessment outcomes improves generalisability and how this can be applied by supervisors and academic staff in assessment. I think the statistical approach taken in rigorous it just needs to justified more clearly. For example, with research question one it is not quite clear from the paper why we need to distil the APP down to one of two factors. This question also mentions clinical performance, but the other element explored is professionalism.

Abstract:

Abstract is well structured. I know there it is difficult with a restricted word count but the reason for separating out the 2 dimensions could be explained a little more. It is not clear if this was the intention during the original design of the APP or done for the purpose of this study.

Introduction:

Page 7, 2nd sentence is long and compound. Could this be broken up for clarity as these are important foundational point.

The next sentence beginning ‘the authentic workplace’ needs context, it is no clear what when consider holistically refers to.

The next sentence also needs contextualisation, the practice development crucible needs explanation. It alludes to four key intersecting influences but does not go on to explain what these are. I am assuming they are contained in the next sentence, but this is not clear.

My opinion, but I find the use of frequent use of bracketed examples (eg…) distracting to read. If these were removed and the example incorporated into the text, this would improve flow.

Pg 7: the sentence beginning “Nevertheless, these independent…” needs further simplification of terms. Concepts such as multidimensional and holistic have not been established in the context of this paper.

Last sentence of page 7 can be simplified – I don’t think this needs to be positioned within a sociocultural framework, it just is what it is – or explain what is meant in this context by a socio-cultural framework.

Second sentence of page 8 needs referencing

Next sentence beginning with “sampling process” I would disagree it is a sampling process, it is just an assessment process. I would also disagree it considers “the evolving and dynamic nature of complex workplaces” – I think there needs to be careful consideration of simplifying the language in this intro as it can imply meaning that is not intended or needs to be contextualised.

I think the main paragraph on page 9 is a bit superfluous, The first part about generalisations can be simplified into a research aim and the justified. So can the part about the research collaboration, it is currently somewhat overstated. It is great that the research was a collaboration with leaders in this area, but this is evident from the list of authors and in a study of this nature has little bearing on the findings.

Methods

The methods section is well written.

The outcome measure section does not mention the global rating scale on the APP which is often used as an overall indication of student performance and in some cases represents 20% of the overall student mark. Why was this not included, especially given it may be highly relevant to research question 1.

Results and Discussion

Please see general comments above as these are relevant to the findings and discussion section.

There appears to be some new information presented in the discussion for example sections of the paragraph that begins on page 17. This should be reviewed.

6. PLOS authors have the option to publish the peer review history of their article (what does this mean? ). If published, this will include your full peer review and any attached files.

**Do you want your identity to be public for this peer review?** For information about this choice, including consent withdrawal, please see our Privacy Policy .

Reviewer #1: **Yes: ** Luke Wakely

---

## [Author Response · Author response to Decision Letter 1]

5 Dec 2024

Please see attached response to reviewers document, as well as the revised manuscript (with and without changes highlighted). We hope we have adequately addressed the editor/reviewers comments and look forward to hearing back in relation to these.

---

## [Decision Letter · Decision Letter 1]

19 Dec 2024

PONE-D-24-38498R1The Assessment of Physiotherapy Practice is a robust measure of entry-level physiotherapy standards: Reliability and validity evidence from a large, representative samplePLOS ONE

Dear Dr. Reubenson,

Thank you for submitting your manuscript to PLOS ONE. After careful consideration, we feel that it has merit but does not fully meet PLOS ONE’s publication criteria as it currently stands. Therefore, we invite you to submit a revised version of the manuscript that addresses the points raised during the review process.

We look forward to receiving your revised manuscript.

Kind regards,

Henri Tilga, PhD

Academic Editor

PLOS ONE

Journal Requirements:

Reviewers' comments:

Reviewer's Responses to Questions

**Comments to the Author**

1. If the authors have adequately addressed your comments raised in a previous round of review and you feel that this manuscript is now acceptable for publication, you may indicate that here to bypass the “Comments to the Author” section, enter your conflict of interest statement in the “Confidential to Editor” section, and submit your "Accept" recommendation.

Reviewer #2: All comments have been addressed

2. Is the manuscript technically sound, and do the data support the conclusions?

Reviewer #2: Partly

3. Has the statistical analysis been performed appropriately and rigorously? 

Reviewer #2: No

4. Have the authors made all data underlying the findings in their manuscript fully available?

Reviewer #2: No

5. Is the manuscript presented in an intelligible fashion and written in standard English?

Reviewer #2: Yes

6. Review Comments to the Author

Reviewer #2: The present manuscript deals with the study of the APP measure, a 20-item instrument used to assess entry-level physiotherapy practice. I have been asked to review this manuscript and have seen that it already was revised. I have decided not to read comments raised in earlier reviews to evaluate the manuscript free from others' perspectives.

Overall, the manuscript is well written and interesting. I applaud the authors for collecting such a diverse and large data set and providing interesting and well-done analyses. However, my reading indicated some points that should be addressed for publication, and I expect that some points might affect the findings (foremost the estimation method of factor analyses).

(1) Authors use ML estimation for the factor analyses, but the measure uses a 4-point rating scale. However, there is a discrepancy between the assumptions of ML (normality and continuous data distribution) that are violated by such response scales. This leads to biases and misestimations. Accordingly, the WLSMV estimator has been introduced to deal with such issues (see Brauer et al., 2023; for a discussion). The minimum requirement is that the data are re-analyzed with the WLSMV estimator and findings are provided to examine the degree of change that is caused by the estimator.

Brauer, K., Ranger, J., & Ziegler, M. (2023). Confirmatory factor analyses in Psychological Test Adaptation and Development: A non-technical discussion of the WLSMV estimator. Psychological Test Adaptation and Development, 4, 4-12. https://doi.org/10.1027/2698-1866/a000034

(2) The evaluation of loadings is very lenient. I recommend to use a clear cut-off criterion to evaluate when a loading is substantial. The literature frequently refers .32 as a meaning loading. Adopting this approach would eliminate Item 4 (maybe also Item 2). However, I think a re-evaluation of the loadings should be done on basis of the results when using the WLSMV estimator.

(3) I have missed information about the reliability (McDonald's omega and retest-correlations over time). Since the data are available, these important findings should be provided.

(4) I have not found the items in verbatim. Please report them to allow others to use the instrument.

I hope that these comments contribute in publishing the manuscript. I am looking forward to read the updated version of the manuscript.

7. PLOS authors have the option to publish the peer review history of their article (what does this mean? ). If published, this will include your full peer review and any attached files.

**Do you want your identity to be public for this peer review?** For information about this choice, including consent withdrawal, please see our Privacy Policy .

Reviewer #2: No

---

## [Author Response · Author response to Decision Letter 2]

15 Jan 2025

Please see word document attached with this submission.

---

## [Decision Letter · Decision Letter 2]

16 Jan 2025

PONE-D-24-38498R2The Assessment of Physiotherapy Practice is a robust measure of entry-level physiotherapy standards: Reliability and validity evidence from a large, representative samplePLOS ONE

Dear Dr. Reubenson,

Thank you for submitting your manuscript to PLOS ONE. After careful consideration, we feel that it has merit but does not fully meet PLOS ONE’s publication criteria as it currently stands. Therefore, we invite you to submit a revised version of the manuscript that addresses the points raised during the review process.

We look forward to receiving your revised manuscript.

Kind regards,

Henri Tilga, PhD

Academic Editor

PLOS ONE

Reviewers' comments:

Reviewer's Responses to Questions

**Comments to the Author**

1. If the authors have adequately addressed your comments raised in a previous round of review and you feel that this manuscript is now acceptable for publication, you may indicate that here to bypass the “Comments to the Author” section, enter your conflict of interest statement in the “Confidential to Editor” section, and submit your "Accept" recommendation.

Reviewer #2: All comments have been addressed

2. Is the manuscript technically sound, and do the data support the conclusions?

Reviewer #2: (No Response)

3. Has the statistical analysis been performed appropriately and rigorously? 

Reviewer #2: (No Response)

4. Have the authors made all data underlying the findings in their manuscript fully available?

Reviewer #2: No

5. Is the manuscript presented in an intelligible fashion and written in standard English?

Reviewer #2: Yes

6. Review Comments to the Author

Reviewer #2: Thank you for addressing my comments. I applaud the efforts made in the revisions.

In response to my comment #2, which raises concerns about the interpretation of the loadings, you ask why I raised this point. First, using the widely used cut-off of .32 allows to identify cross-loadings several cross-loadings in your scale that are not detected with a comparatively lenient cut-off of .40. Secondly, you do not present or discuss consequences that follow from the loadings not meeting cut-offs or showing substantial cross-loadings. The standard would be to exclude items that do either not load on the respective factor or show cross-loadings and compute a new factor model, which is then evaluated regarding fit and loadings. This is done until a final well-defined model is available. The discussion should then highlight which items were excluded and why, and what the final version of the measurement instrument is.

7. PLOS authors have the option to publish the peer review history of their article (what does this mean? ). If published, this will include your full peer review and any attached files.

**Do you want your identity to be public for this peer review?** For information about this choice, including consent withdrawal, please see our Privacy Policy .

Reviewer #2: No

---

## [Author Response · Author response to Decision Letter 3]

20 Jan 2025

Please see attached 'response to reviewers' document.

---

## [Decision Letter · Decision Letter 3]

3 Mar 2025

PONE-D-24-38498R3The Assessment of Physiotherapy Practice is a robust measure of entry-level physiotherapy standards: Reliability and validity evidence from a large, representative samplePLOS ONE

Dear Dr. Reubenson,

Thank you for submitting your manuscript to PLOS ONE. After careful consideration, we feel that it has merit but does not fully meet PLOS ONE’s publication criteria as it currently stands. Therefore, we invite you to submit a revised version of the manuscript that addresses the points raised during the review process.

We look forward to receiving your revised manuscript.

Kind regards,

Mansour Abdullah Alshehri

Academic Editor

PLOS ONE

Journal Requirements:

Reviewers' comments:

Reviewer's Responses to Questions

**Comments to the Author**

1. If the authors have adequately addressed your comments raised in a previous round of review and you feel that this manuscript is now acceptable for publication, you may indicate that here to bypass the “Comments to the Author” section, enter your conflict of interest statement in the “Confidential to Editor” section, and submit your "Accept" recommendation.

Reviewer #2: All comments have been addressed

Reviewer #3: (No Response)

Reviewer #4: (No Response)

2. Is the manuscript technically sound, and do the data support the conclusions?

Reviewer #2: Partly

Reviewer #3: Yes

Reviewer #4: Yes

3. Has the statistical analysis been performed appropriately and rigorously? 

Reviewer #2: No

Reviewer #3: Yes

Reviewer #4: Yes

4. Have the authors made all data underlying the findings in their manuscript fully available?

Reviewer #2: Yes

Reviewer #3: Yes

Reviewer #4: Yes

5. Is the manuscript presented in an intelligible fashion and written in standard English?

Reviewer #2: (No Response)

Reviewer #3: Yes

Reviewer #4: Yes

6. Review Comments to the Author

Reviewer #2: (No Response)

Reviewer #3: Dear authors,

First, I would like to thank you for the opportunity to review your work. It addresses an extremely important aspect of assessing access levels to the Physiotherapy profession and the quality of competency acquisition assessment processes. The example provided should be adopted in several countries that do not use this evaluation scale.

Congratulations! I believe the responses given to the different reviewers' suggestions were well executed, and I find the article very well developed.

Reviewer #4: Overall, the manuscript is technically sound

You have presented a well-structured and detailed response that shows that you thoroughly considered the feedback from of Reviewer #2. You have clarified your rationale for the cut-offs you selected for factor loading, described how you dealt with cross-loadings and included a plan for refining ambiguous items as a future research area. I support your approach of balancing between statistical rigor and consideration for conceptual validity.

However, to make your argument stronger, you may discuss whether you considered Reviewer #2’s suggestion to recompute the factor model iteratively until stability was reached, how that looked like and why you eventually stuck to the current approach. In addition to this, you could provide a clearer framework for deciding to keep items with cross loadings. This could be for example through specifying a threshold or guideline that weighs conceptual importance over statistical considerations. This would be very helpful for other researchers seeking to replicate your approach.

I recommend a minor revision for these refinements, which will enhance clarity and transparency without requiring substantial changes, given the strength of the current version of your manuscript.

7. PLOS authors have the option to publish the peer review history of their article (what does this mean? ). If published, this will include your full peer review and any attached files.

**Do you want your identity to be public for this peer review?** For information about this choice, including consent withdrawal, please see our Privacy Policy .

Reviewer #2: No

Reviewer #3: **Yes: ** Tiago Atalaia

Reviewer #4: No

---

## [Author Response · Author response to Decision Letter 4]

4 Mar 2025

Please see response letter attached with this submission.

---

## [Editor Report · Decision Letter 4]

6 Mar 2025

The Assessment of Physiotherapy Practice is a robust measure of entry-level physiotherapy standards: Reliability and validity evidence from a large, representative sample

PONE-D-24-38498R4

Dear Dr. Alan,

We’re pleased to inform you that your manuscript has been judged scientifically suitable for publication and will be formally accepted for publication once it meets all outstanding technical requirements.

Kind regards,

Mansour Abdullah Alshehri

Academic Editor

PLOS ONE
---

## [Editor Report · Acceptance letter]

PONE-D-24-38498R4

PLOS ONE

Dear Dr. Reubenson,

I'm pleased to inform you that your manuscript has been deemed suitable for publication in PLOS ONE. Congratulations! Your manuscript is now being handed over to our production team.

Kind regards,

on behalf of

Dr. Mansour Abdullah Alshehri

Academic Editor

PLOS ONE